# Radiation-Induced Intestinal Normal Tissue Toxicity: Implications for Altered Proteome Profile

**DOI:** 10.3390/genes13112006

**Published:** 2022-11-02

**Authors:** Enoch K. Larrey, Rupak Pathak

**Affiliations:** 1Division of Radiation Health, Department of Pharmaceutical Sciences, College of Pharmacy, University of Arkansas for Medical Sciences, 4301 W. Markham St., Little Rock, AR 72205, USA; 2Department of Information Science, University of Arkansas at Little Rock, 2801 S University Ave, Little Rock, AR 72204, USA

**Keywords:** radiation, enteropathy, proteomics, intestine, biomarkers

## Abstract

Radiation-induced toxicity to healthy/normal intestinal tissues, especially during radiotherapy, limits the radiation dose necessary to effectively eradicate tumors of the abdomen and pelvis. Although the pathogenesis of intestinal radiation toxicity is highly complex, understanding post-irradiation alterations in protein profiles can provide crucial insights that make radiotherapy safer and more efficient and allow for increasing the radiation dose during cancer treatment. Recent preclinical and clinical studies have advanced our current understanding of the molecular changes associated with radiation-induced intestinal damage by assessing changes in protein expression with mass spectrometry-based approaches and 2-dimensional difference gel electrophoresis. Studies by various groups have demonstrated that proteins that are involved in the inflammatory response, the apoptotic pathway, reactive oxygen species scavenging, and cell proliferation can be targeted to develop effective radiation countermeasures. Moreover, altered protein profiles serve as a crucial biomarkers for intestinal radiation damage. In this review, we present alterations in protein signatures following intestinal radiation damage as detected by proteomics approaches in preclinical and clinical models with the aim of providing a better understanding of how to accomplish intestinal protection against radiation damage.

## 1. Introduction

The intestine is one of the most radiation-sensitive organs, and the risk of damage to healthy/normal tissues surrounding a tumor that is targeted by radiotherapy is very high. Intestinal toxicity caused by radiotherapy, commonly known as radiation enteropathy [1,2,3], is a major but largely unaddressed problem for ~3.4 million radiation-treated cancer survivors in the United States (US). Despite recent progress in radiation delivery techniques, radiation enteropathy is a major dose-limiting factor during abdominal and pelvic radiation therapy [1,4,5]. This off-target damage is unavoidable due to the highly penetrating nature and uniform dose deposition per unit distance of X-rays and γ-rays, largely used in external beam radiation therapy, which is delivered in multiple fractions in cancer clinics. Annually, more than 200,000 US patients with cancer receive fractionated radiation to the abdomen or pelvis [5,6].

Radiation enteropathy symptoms can appear within hours (early), weeks (acute), or even months or years (delayed or chronic) after radiotherapy [1]. Following radiotherapy, 60% to 80% of patients commonly develop acute clinical and histopathologic symptoms (e.g., diarrhea, abdominal pain, bloody stool, nausea, inflammatory cell infiltration, reduced crypt cell mitosis, and mucosal epithelial denudation and ulceration) [1,5]. While most acute symptoms resolve within 1 to 3 months after radiotherapy, about 90% of these patients suffer from permanent changes in bowel habits (particularly dysmotility and malabsorption). Of these, 50% experience substantial decreases in quality of life, and 10% to 15% develop life-threatening complications within 10 to 20 years after radiotherapy (e.g., submucosal fibrosis, vascular sclerosis, secondary cancer, bowel obstruction, and tissue necrosis) [1,5]. The chronic symptoms tend to be irreversible and progressive and generally have a poor prognosis [6]. Surgery is the preferred treatment option, but the mortality rate is high. Although chronic radiation enteropathy is more prevalent than inflammatory bowel disease [1], radiation enteropathy draws less attention because of false perceptions that it is not prevalent, the lack of multidisciplinary expertise required to treat these patients, and the absence of consensus on the treatment success rate [1].

The pathogenesis of radiation enteropathy is not fully understood. It involves a complex network of cellular death, proliferation arrest, and/or activation. Acute radiation enteropathy involves apoptotic or mitotic loss of intestinal epithelial stem and transiently amplifying cells in the crypt region [7]. Crypt damage impairs the replacement of the mucosal epithelial surface, blunting villus height and diminishing barrier integrity. A compromised mucosal barrier impairs absorption and facilitates the invasion of luminal intestinal bacteria into the systemic circulation, increasing the risk of inflammation and sepsis. In contrast, chronic radiation enteropathy is thought to result from the proliferative arrest (senescence) of different cell types in the irradiated microenvironment as well as from changes in their function [1,7]. Functional changes may affect the secretion of soluble mediators (e.g., cytokines, chemokines, and growth factors), cell-cell interactions, profiles of cell-surface adhesion molecules, and cell trafficking, all of which are the consequences of altered protein expression [1,6,7].

Proteomics is a powerful tool to identify and quantitate changes in the expression of thousands of proteins from any tissue, including intestinal tissues, after exposure to any stimuli of external or internal origin. Two proteomics approaches are commonly used to estimate the protein abundance in intestinal tissues. One approach is gel-based and uses either 2-dimensional difference gel electrophoresis (2D-DIGE) or 2-dimensional gel electrophoresis (2-DE), and the other is mass spectrometry (MS). Figure 1 shows a typical flow chart of two different proteomic analysis methods such as gel-based and mass spectrometry-based methods that can be used to assess intestinal radiation toxicity using preclinical and clinical models. These quantitative methods allow the estimation of the relative abundance and identification of the proteome in the intestinal cells in culture and in the intestinal tissue after total body irradiation (TBI) or partial body irradiation (PBI) and in plasma samples after targeted radiation of the abdominal region. These quantitative proteomics approaches continue to provide useful insights into the molecular mechanism of radiation-induced intestinal damage. This review article summarizes the changes in protein profiles in the intestinal cells in culture, intestinal tissues in rodents and nonhuman primates following radiation, or in patients undergoing abdominopelvic radiotherapy. The aim of this review is to identify alterations in the intestinal protein landscape, which may help to target signaling pathways in order to minimize intestinal radiation toxicity.

## 2. Proteomics Methods Used to Separate Proteins 

Gel- and MS-based methods are the 2 most popular proteomics methods used to separate proteins for protein identification, quantitation, localization, and assessing post-translational modification and functional and structural changes in proteins.

### 2.1. Gel-Based Proteomics Method

2-DE, which uses an electric current to separate proteins in a gel based on their charge (1st dimension) and mass (2nd dimension), was the first proteomic technique developed [8]. 2-DE is also known as differential display proteomics or expression proteomics because it permits the analysis of differentially expressed proteins under specific conditions in a targeted manner. However, with 2-DE, only one sample can be run per gel. 2D-DIGE was developed to overcome this limitation. 2D-DIGE uses different covalently tagged fluorescent dyes that allow the simultaneous comparison of 2 to 3 protein samples on the same gel without compromising the migration of proteins [9]. These gel-based methods are used to separate proteins before further analysis by other methods (e.g., MS), as well as for relative expression profiling. Figure 2 illustrates both 2-DE and 2D-DIGE workflow, adapted from Cristea et al. [10].

### 2.2. Mass Spectrometry-Based Proteomics

There are several “gel-free” methods for separating proteins, such as MS-based methods. MS was invented more than a century ago; however, the technique has experienced a recent surge in usage in proteomics [11]. The application of MS in protein studies involving matrix-assisted laser desorption ionization and time-of-flight MS in early 1990s advanced the field to a great extent [12,13,14]. Since then, efforts have been made to enhance its accuracy, sensitivity, and dynamic range [15,16]. Today, a single MS-based proteomics operation can be used to estimate the absolute and relative abundance of all proteins in a given cell or tissue with high sensitivity and throughput compared to other methods. Moreover, the same detection method is applicable to different samples (blood, serum, or tissue) with no limitation in sensitivity or specificity [17]. A classical bottom-up workflow in MS-based proteomics involves protein digestion into peptides, liquid chromatography separation, measurement of peptides using tandem MS, and subsequent database searching to use the information on all known peptides in the database for the assignment of peptides to proteins. This strategy makes it possible to identify proteins in a complex mixture [18] and is now an integral part of MS-based proteomics due to its robustness and accuracy [15]. There are multiple variations of the MS method with modifications made to the various steps in the typical bottom-up proteomics workflow to accomplish specific goals. For example, while MS proteomics can be used for label-free quantitation of spectra, peptides can also be labeled with isotopes or chemicals during sample preparation in order to estimate the absolute and relative abundance of proteins in complex environments. Methods requiring labelling of peptides include stable isotope labelling by amino acids in cell culture (SILAC) [19], tandem mass tags (TMTs) [20,21], and isobaric tags for relative and absolute quantification (iTRAQ) [22]. Although SILAC generates high-quality quantitative data, culturing of cells in a medium containing isotopes is a requirement and also a weakness of the method. This limitation has been addressed by the invention of super-SILAC, which permits quantitative proteomic analysis of human tissue samples by mixing SILAC-labelled cell lines with human tissues [23,24,25]. On the other hand, in chemical labeling approaches, such as iTRAQ and TMT, isobaric tags are directly added to enzyme-digested peptides without the need for labeling in an isotope-containing culture medium. For instance, using TMT, 54 samples can be tagged with different combinations of isobaric tags and analyzed at the same time [26,27,28], making it possible to compare the abundances and modifications in several samples without pretreatment [29]. In the MS method, data is generated by 3 mechanisms, namely, data-dependent acquisition (DDA), data-independent acquisition (DIA), and targeted MS. The differentiating factor between DDA and DIA is in the charge usage. While DDA uses a selected number of peptide fragments and their estimated mass-to-charge ratio, the DIA approach requires precursor ions of all peptide fragments that fall within a particular threshold in order to maximize coverage. Targeted MS is a more focused approach, as the name suggests, and deals with monitoring peptides or proteins of interest during chromatography until an appreciable number of the peptide fragments is obtained for maximum coverage [11]. Figure 3 illustrates a typical MS workflow adapted from Bartke et al. [30].

## 3. Radiation-Induced Change in Protein Profile in Intestinal Cells or Tissues

### 3.1. Radiation Alters Protein Profiles in Intestinal Cells in Culture

An in vitro proteomics study with rat small intestinal epithelial cells (IEC-6) revealed that a single exposure to 25 Gy ^60^Co-γ-rays differentially altered 16 proteins compared to sham-irradiated cells at 24 h [31]. The proteomics data also indicated that radiation-induced differentially expressed proteins in IEC-6 cells are involved in the cellular processes of anti-oxidation, structural development, metabolism, and protein post-translational modifications [31]. Further confirmation of proteomics data with immunoblot analysis demonstrated a significant reduction in the expression of stress-70 protein (also known as GRP75) in IEC-6 cells after radiation [31]. This proteome analysis may contribute to the elucidation of a molecular mechanism of radiation damage in intestinal cells.

### 3.2. Radiation Alters Protein Profile in the Intestinal Tissue of Rodents

Proteomics studies of intestinal tissue following irradiation have been extensively investigated, primarily in mice and rats. Table 1 shows preclinical models used for the quantitative proteomics study of radiation damage to normal gastrointestinal tissues and the identification of radiation-responsive proteins. Notably, protein profile changes following irradiation depend on animal strain, sex, and age; radiation dose and type; post-irradiation time interval; and the radiation delivery technique. For example, a proteomics study of intestinal tissues of 6- to 8-week-old male C57BL/6 mice at 1 h after exposure to 9 Gy γ-rays found 17 proteins were expressed only in the irradiated group compared to the unirradiated control group [32]. The dysregulated proteins were involved in biological roles, including post-translational modifications, protein turnover and chaperones, bimolecular transportation and metabolism, cytoskeletal structure, energy production and conversion, and signal transduction mechanisms [32]. Significantly, MYC transcription factor was identified as the only upstream regulator affected by radiation exposure [32]. With the help of commercially available antibody kits, the abnormal expression of ATP synthase subunit D, aldehyde dehydrogenase, Cox5a, CRP, multifaceted C1qbp, Oat, and Pcna was confirmed after irradiation [32]. According to Zhang et al., in comparison to an average of 638 ± 39 protein spots identified by 2-DE in sham-irradiation, exposure to 9 Gy TBI yielded an average of 566 ± 32 protein spots at 3 h and 591 ± 29 at 72 h in the intestinal tissue of 58- to 62-day-old male BALB/c mice [33]. Further analysis by peptide mass fingerprinting revealed that 19 proteins were differentially expressed following irradiation. Proteins involved in redox regulation, such as peroxiredoxin I and glutathione S-transferase P2, were upregulated, while antioxidant protein 2 (also known as 1-Cys peroxiredoxin) was downregulated [33]. Notably, expression of enolase, a glycolytic enzyme, was upregulated in the intestine 3 h after 9 Gy TBI but not at 72 h when compared to sham-irradiated controls [33]. Another study by Lim et al. showed that 1 Gy TBI exposure in 7-week-old C57BL/6 mice resulted in 49 differential intestinal protein spots out of a total of 977 spots analyzed by 2-DE for both irradiated and unirradiated groups [34]. Only 5 of the 49 differential spots were identified and verified by commercially available antibodies [34]. Importantly, the expression of phosphoglycerate kinase 1 (PGK1) was significantly higher at 24 h after 1 Gy TBI in mouse intestinal tissue [34]. Rosen et al. demonstrated that 6- to 8-week-old male CD2F1 mice irradiated with 11 Gy γ-rays exhibited 26 differentially altered proteins in the intestine compared to sham-irradiated controls 24 h after exposure [35]; however, the expression of 13 of these proteins was normalized in mice groups treated with either of 2 vitamin E family members, γ-tocotrienol or tocopherol succinate, prior to radiation exposure [35]. Another intestinal proteomics study conducted in 8- to 10-week-old male C57BL/6J mice exposed to a single uniform X-ray TBI dose of 8, 10, 12, or 14 Gy reported that the expression of 103 proteins was consistently altered 1, 3, and 6 days after exposure [36]. Of these, 46 proteins were consistently activated, and 57 proteins were consistently inhibited [36]. Further analysis showed that due to the consistent alteration in protein expression over the dose range and days, molecular functions associated with thiol ester hydrolase activity, serine-type endopeptidase activity, nucleosomal DNA binding, acyl-CoA hydrolase activity, palmitoyl-CoA hydrolase activity, and carboxylic ester hydrolase activity were significantly elevated, whereas molecular function associated with poly(A) RNA binding was significantly reduced [36]. Furthermore, it was revealed that 16 of the proteins were significantly associated with retinoic acid, including Aldh1a1, Apoa2, Apoe, Rbp2, Rdh7, Ttr, and a series of Akr proteins; 3 proteins were connected to radiation (Eef1d, Ptprc, and Sod2); and 40 proteins were associated with the inflammation [36]. The authors also indicated the expression of 44 upstream regulators was impacted by TBI irradiation [36]. Finally, the authors showed a dose dependent increase in 5 proteins (such as FABP1, FGA, FGB, FGG, and HP) in the intestinal sample of C57BL/6 male mice on day 3 following exposure to 8, 10, 12, and 14 Gy TBI, which could be used as radiation biodosimetric markers for intestinal injury [36]. Han et al. analyzed the intestinal proteome of 6-week-old female C57BL/6J mice exposed to 7 Gy γ-rays TBI 10 days after irradiation and identified 186 proteins in the control group, 270 proteins in the irradiated group, and 238 proteins in a group irradiated and treated with human placenta-derived stem cells (hPDSCs) [37]. Further analysis revealed 68 uniquely expressed proteins in the irradiated only mice compared to 27 in the control group and 38 in the radiation plus hPDSCs-treated group [37]. IL-10, glycoprotein, TIMP-1, and antioxidant protein, GST mu type 1, which decreased in expression following radiation exposure, were verified by immunoblot; however, the administration of hPDSCs increased the expression of these proteins [37]. Moreover, Wang et al. examined intestinal tissue of male C57BL/6J mice 3 days after 13 Gy ^137^Cs abdominal γ-ray irradiation and identified 1279 proteins that were differentially expressed compared to sham-irradiated controls [38].

Two proteomics studies of intestinal tissues have utilized abdominal irradiation of male Sprague–Dawley rats. In one of these studies, intestinal tissues of adult rats were examined 4 days after exposure to 10 Gy γ-rays, revealing 86 differentially expressed proteins involved in lipid, protein, carbohydrate, and other metabolic and cellular processes [39]. In the second study, 6-week-old rats were exposed to 20 Gy X-rays, and 10 weeks after exposure, 6692 proteins were identified, of which 5756 were quantified [40]. Of the quantified proteins, 320 were significantly altered [40]. The differentially expressed proteins were involved primarily in biological regulation and single-organism processes, metabolic processes, and the response to stimulus [40]. Validation by parallel reaction monitoring showed that proteins associated with complement and coagulation cascades (FGG, C3, and F2), regulation of the actin cytoskeleton (F2, ITGB2, and ITGAM), and leukocyte trans-endothelial migration (CYBB, ITGB2, and ITGAM) were upregulated [40]. The findings of these studies establish the fact that exposure to ionizing radiation alters the intestinal protein expression profile.

### 3.3. Radiation Alters Protein Profile in the Intestinal Tissue of Nonhuman Primates 

PBI with minimal bone marrow sparing (2.5% to 5% sparing) causes a change in the intestinal proteome landscape of nonhuman primates. Huang et al. exposed male rhesus macaques (*Macaca mulatta*) to 12 Gy PBI with 2.5% bone marrow sparing (PBI/BM2.5) using a 6 MV linear accelerator and harvested intestinal tissues on days 4, 8/9, 11/12, 15, and 21/22 [41]. Out of the 3700 proteins that were identified, 3245 were quantified [41]. Notably, PBI/BM2.5 altered the expression of 289 proteins significantly and consistently across at least 3 time points, of which 263 proteins were upregulated while 26 proteins were downregulated [41]. Further analysis revealed that 18 upstream regulators were significantly dysregulated, out of which 15 were upregulated and 3 were downregulated [41]. In addition, the authors observed a strong positive correlation between downregulated proteins and reduction in crypt number [41]. Finally, the authors found that inflammatory proteins such as ACTA1, DUOX2, DNM1, COL6A3, GAL, HP, and S100A8 were significantly and consistently upregulated, while many proteins related to retinoic acid activity, such as retinal dehydrogenase and retinal reductase, were downregulated following PBI/BM2.5 [41]. 

## 4. Therapeutic Radiation Alters the Plasma Protein Profile of Rectal Cancer Patients

Radiation is an integral part of cancer treatment. Approximately 50% of the patients with cancer in the US, including those suffering from abdominopelvic cancer, receive radiation therapy at a certain stage of their cancer treatment. One of the major detrimental side effects of abdominopelvic radiation therapy is intestinal healthy tissue toxicity, which may alter plasma protein profile. A study by Holm et al. of patients with stage II and III rectal cancer who were rectally exposed to 5 fractions of 5 Gy radiation for 5 consecutive days demonstrated significant alteration of 14 plasma proteins in stage II patients and 28 in stage III patients compared to rectal cancer patients who did not receive radiation therapy [42]. Interestingly, in stage II patients, all 14 altered plasma proteins were downregulated, while in stage III patients, all 28 altered proteins were upregulated following radiation therapy compared to rectal cancer patients who did not receive radiation therapy [42]. The significantly downregulated proteins in stage II patients include hemoglobin subunit beta, hemoglobin subunit alpha, and lysozyme C, while the highly upregulated proteins in stage III patients were serum amyloid A1 and synapsin 2 [42]. This study indicates that cancer stage modulates plasma protein levels in radiation-induced intestinal damage following radiotherapy. 

## 5. Signaling Pathways Altered as a Result of Intestinal Radiation Toxicity 

Systematic studies of altered signaling pathways in the intestines following radiation damage may provide crucial information to develop countermeasure strategies. A number of preclinical studies with murine models of intestinal radiation toxicity exhibit dysregulation of various pathways in the intestinal tissue. For example, the proteasome and protein processing pathways were significantly altered in the intestinal tissue samples of 6- to 8-week-old male C57BL/6J mice exposed to 9 Gy γ-rays at 1 h post exposure [32]. Another study demonstrated alterations in 5 canonical signaling pathways such as Rho family GTPases, glycolysis I, xenobiotic metabolism, 14-3-3-mediated signaling, and retinol biosynthesis in the intestinal tissues of male CD2F1 mice 24 h after exposure to 11 Gy TBI [35]. Huang et al. demonstrated dysregulation in protein kinase A signaling, acute phase response signaling, and LXR/RXR signaling in the intestinal tissue of 8- to 10-week-old male C57BL/6J mice following exposure to TBI X-rays in the range of 8 to 14 Gy at 1, 3, and 6 d after exposure [36]. Finally, DNA damage and apoptotic pathways were shown to alter in the intestinal tissue of mice on day 3 following 13 Gy abdominal irradiation with γ-rays [38]. 

Preclinical studies with rat models of intestinal radiation toxicity also exhibit that radiation dysregulates a number of intestinal signaling pathways. FAS and glycolysis signaling pathways were dysregulated in the intestinal tissues of Sprague–Dawley rats following 10 Gy abdominal γ-ray irradiation on day 4 after exposure [39]. A study of 6-week-old male Sprague–Dawley rats exposed to 20 Gy abdominal X-ray irradiation exhibited dysregulation in complement and coagulation cascades, amoebiasis, phagosome, lysosome, focal adhesion, oxytocin signaling in the intestinal tissue 10 weeks after exposure [40].

In addition, radiation was shown to affect various signaling pathways in nonhuman primates. For example, 4 canonical pathways—GP6 signaling pathway, acute phase response signaling, LXR/RXR activation, and intrinsic prothrombin activation pathway—were dysregulated in the intestinal tissue of nonhuman primates following 12 Gy PBI/2.5 BM at various time points ranging from 4 to 22 days [41]. Table 2 shows different pathways affected in intestinal tissues following irradiation.

## 6. Discussion

In this review, we have discussed the importance of studying intestinal proteomics in regard to radiation toxicity, provided an overview of methods used for proteomics studies, and summarized the total number of altered proteins following intestinal radiation injury as well as the major intestinal pathways dysregulated after radiation damage. Studies of intestinal proteomics following radiation injury may help to identify crucial protein biomarkers that can be targeted to develop new countermeasure strategies to limit acute and delayed intestinal toxicity. Limiting intestinal radiation toxicity would not only improve the quality of life of radiation victims or radiation-treated cancer survivors who are suffering from intestinal toxicity, but it also would significantly reduce the burden on health care systems.

Identifying protein biomarkers by proteomics approaches is a challenging task as protein profiles change based on a number of factors, including animal strain, age, species, and sex; radiation dose, quality and mode of delivery; and post-irradiation time points. For example, 17 proteins were differentially expressed in male C57BL/6 mice after 1 h of 9 Gy TBI with γ-rays, whereas 49 proteins were differentially expressed in female C57BL/6 mice 24 h after 1 Gy TBI with γ-rays as compared to sham-irradiated control groups [32,34], suggesting radiation dose, animal sex, and post-irradiation times are confounding factors in altering protein level in the intestinal tissue. Moreover, a comparison of 2 other intestinal proteomics studies, where the radiation delivery technique (TBI) and radiation type (γ-rays) were the same, revealed that a total of 26 proteins were altered in male CD2F1 mice 24 h after 11 Gy irradiation, while 95 proteins were differentially expressed 10 days after 7 Gy irradiation in female C57BL/6 J mice compared to the respective sham-irradiated controls [35,37]. Similarly, in Sprague–Dawley rats, 86 intestinal proteins were altered on day 4 after 10 Gy γ-ray, and 320 intestinal proteins were altered at 10 weeks after 20 Gy X-ray irradiation [39,41]. These results further confirm the challenges faced in identifying biomarkers in proteomics studies. However, with the advancement of protein separation techniques, the proteins in a sample of interest can be characterized more accurately. 

Because it is not practical to collect intestinal samples for proteomics studies from patients accidentally exposed to radiation or patients with cancer being treated with radiation, identifying protein biomarkers by systematically scanning the changes in protein profiles from plasma samples may serve as an alternative approach for predicting the extent of intestinal radiation damage. Plasma collection is relatively convenient and is a less invasive method. A number of studies have shown that after intestinal radiation damage, the expression of the same proteins altered in the plasma sample as well as in the intestinal tissues, suggesting plasma proteins may serve as candidate biomarkers of radiation toxicity to the gastrointestinal tract. For example, serum amyloid A1 SSA1 and c-reactive protein are significantly altered both in plasma samples and intestinal tissues of nonhuman primates following 12 Gy PBI [36,43]. Interestingly, the same heat shock protein, amyloid A1 SSA1, was also altered in the plasma of patients with rectal cancer following 5 fractions of 5 Gy radiation for 5 consecutive days [42]. These findings may provide the impetus for more in-depth research in order to identify plasma protein biomarkers to assess intestinal radiation damage.

All proteins directly or indirectly influence one or more signaling pathways. Therefore, the altered intestinal protein profile following irradiation can be used to predict which pathways are dysregulated. Targeting dysregulated pathways may counteract intestinal radiation injury. In this regard, identifying a common pathway that is dysregulated irrespective of radiation dose and quality, post-irradiation time, and species would be an invaluable target for protection from intestinal radiation damage. Studies have shown that acute phase response signaling and the LXR/RXR pathway are dysregulated both in C57BL/6J following exposure to various doses of TBI with X-rays (8, 10, 12, or 14 Gy) and in rhesus macaques following exposure to 12 Gy PBI with X-rays [36,41]. Similarly, the glycolysis signaling pathway was dysregulated in the intestinal tissue of C57BL/6J exposed to 13 Gy ABI with ^137^Cs γ-rays and in Sprague–Dawley rats exposed to 10 Gy ABI with ^60^Co γ-rays [38,39]. Finally, the coagulation cascade was dysregulated in the intestinal tissue of Sprague–Dawley rats after exposure to 20 Gy ABI with X-rays and in plasma samples of rhesus macaques after exposure to 12 Gy PBI with a 6 MV linear accelerator [40,43].

When the experimental settings (in terms animal species and strains, absorbed dose and dose rate, tissue harvest times, and radiation delivery techniques) are different, causal connections between radiation-induced tissue damage and pathological changes based on proteomics data are particularly difficult to establish. However, the cited literature in this review article indicates that acute effects of radiation cause apoptosis [38], mitochondrial dysfunction [32,37], and altered cell signaling [35]. Radiation causes damage to the mucosal epithelial layer by inducing apoptotic signaling pathways, thereby increasing the risk of luminal microbe infiltration into the circulatory system, resulting in sepsis if treatment does not happen in a timely manner. In addition, radiation downregulates expression of proteins involved in normal mitochondrial activity, such as ATP synthase sub-unit D, Cox5a and antioxidant protein 2 [32,37]. The compromise of mitochondrial activity facilitates generation of excessive reactive oxygen species, leading to oxidative damage of functional activity of several proteins, namely proteasomes [32]. Proteasomes remove damaged proteins which result from radiation-induced tissue damage. To restore mitochondrial function and counteract the excessive ROS generation, glycolytic enzymes (such as enolase and phosphoglycerate kinase 1) [33,34], and ROS scavenging enzymes (such as peroxiredoxin I and glutathione S-transferase P2) [33,35] may be increased, respectively. In addition, radiation inhibits G-protein coupled receptors (GPCRs) from transmission of cell signals because of impairment of structural proteins (such as cytoplasmic actin 2, lamin B1, and ezrin) [35,39], which can adversely affect several cell signaling pathways, including those that play a crucial role in radiation protection. For example, radiation suppresses GPCR-mediated retinoic pathway [35]. A previous study has shown that retinoic acid and TIMP1 prevent radiation-induced apoptosis in endothelial cells [44]. Apoptosis, mitochondrial dysfunction, and altered cell signaling trigger activation of immune cells leading to persistent systemic inflammation. These activated immune cells interact with endothelial cells that form the inner layer of the blood vasculature and transmigrate through the endothelial layer to reach to the damage tissues [40]. Persistent inflammation drives delayed effects of radiation by activation of the complement cascade, enhancing coagulopathy, and fueling thrombotic changes [40,41]. Figure 4 summarizes events and resulting pathological conditions following radiation at both cellular and tissue levels.

In conclusion, data from comprehensive radiation proteomics in the intestinal tissue have deepened the knowledge of molecular mechanisms involved in intestinal radiation toxicity, which holds great promise but needs further validation. Further research and state-of-the-art proteomics have the potential to identify new diagnostic and therapeutic markers to limit radiation-induced intestinal toxicity. In addition, determining the signaling pathways associated with early and delayed reactions of radiation will provide further insight regarding the management of intestinal toxicity.

## Figures and Tables

**Figure 1 genes-13-02006-f001:**
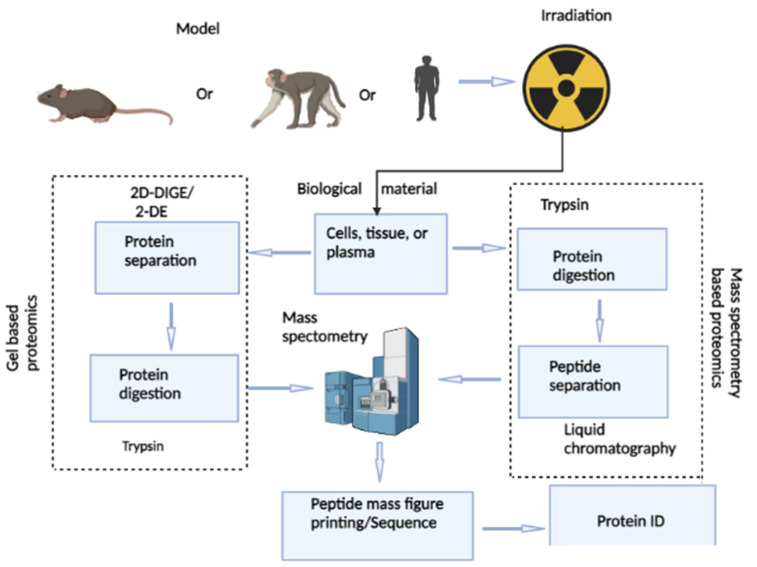
Flow-chart showing the steps involved in intestinal proteomics analysis for gel-based and mass spectrometry-based methods. Briefly, lysates from cells in culture or tissues of rodents and nonhuman primates or plasma from nonhuman primates and humans collected after irradiation are subjected to one of two protein separation methods, gel or mass spectrometry. Subsequently, peptide spectra and corresponding mass to charge ratios are generated by mass spectrometry, searched against protein databases for protein identification and further downstream analysis.

**Figure 2 genes-13-02006-f002:**
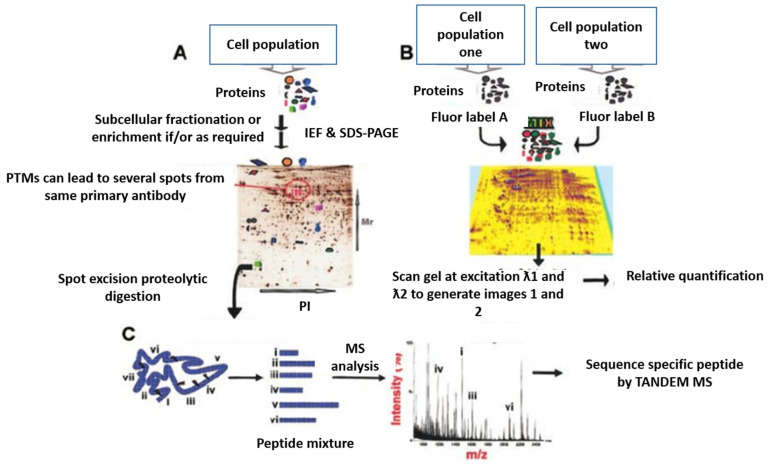
Gel-based proteomics. (**A**). protein separation by 2-DE; (**B**). protein separation by 2-D DIGE; and (**C**). MS for protein identification [10].

**Figure 3 genes-13-02006-f003:**
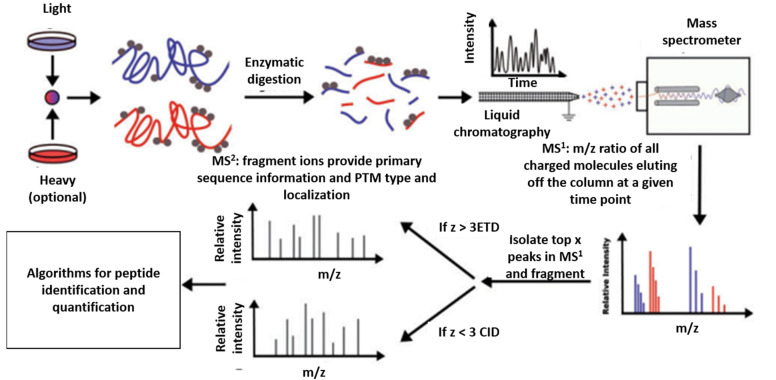
MS-based proteomics. Illustration of the steps involved in MS proteomics analysis [30].

**Figure 4 genes-13-02006-f004:**
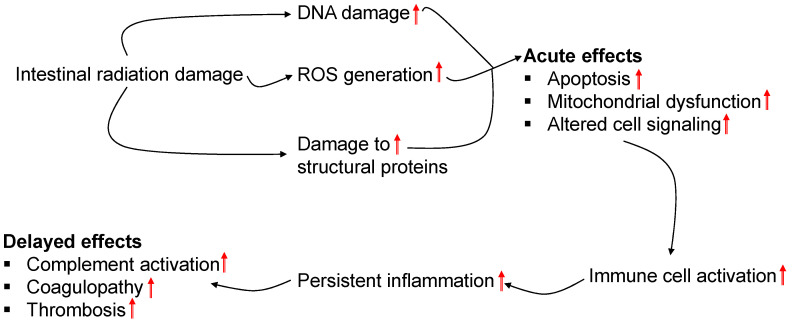
Summary of events and resulting pathological condition because of acute and delayed effects of intestinal radiation injury.

**Table 1 genes-13-02006-t001:** Preclinical rodent models used for quantitative proteomics study in the intestinal tissue following radiation exposure.

Strain	Sex	Age	Tissue Type	Radiation Dose (Gy)	Tissue Harvest Time	Mode of Radiation	Radiation Type	Radiation-Responsive Proteins	Ref.
C57BL/6 mice	Male	6–8 weeks	Jejunum	9	1 h	TBI	γ-rays	ATP synthase subunit D, aldehyde dehydrogenase, Cox 5a, CRP, multifaceted C1qbp, Oat, Pcna	[32]
Bal b/c mice	Male	58–62 days	Intestinal epithelia	9	3 and 72 h	TBI	γ-rays	Enolase and peroxiredoxin	[33]
C57BL/6 mice	Female	7 weeks		1	24 h	TBI	γ-rays	Phosphoglycerate kinase 1	[34]
CD2F1 mice	Male	6–8 weeks	Jejunum	11	24 h	TBI	γ-rays	Cytoplasmic actin 2, dihydropyrimidinaserelated protein 2, ezrin, elongation factor 2, plastin-1, and peroxiredoxin-1	[35]
C57BL/6J mice	Male	8–10 weeks	Ileum	8, 10, 12 and 14	1, 3, and 6 days	TBI	X-rays	Ctsc, Cen pv, amy2, DUOX2, Try4, Fabp1, Dsp	[36]
C57BL/6J mice	Female	6 weeks	Small intestines	7	10 days	TBI	γ-rays	IL-10, glycoprotein, TIMP-1, and antioxidant protein, GST mu type 1	[37]
C57BL/6J mice	Male	6–8 weeks	Small intestines	13	3 days	ABI	γ-rays	Gpx3, Sod3	[38]
Sprague–Dawley rats	Male	Adult	Ileum	10	4 days	ABI	γ-rays	Gelsolin, Prelamin-A/C, Lamin-B1, fructose-bisphosphatealdolase A and α-enolase	[39]
Sprague–Dawley rats	Male	6 weeks	Large intestine	20	10 weeks	ABI	X-rays	FGG, THBS1, AGT, F2, C3, ITGAM, ITGB2, CYBB, QSOX1	[40]

**Table 2 genes-13-02006-t002:** Pathways dysregulated by radiation exposure in normal intestinal tissues.

Strain	Age	Tissue Type	Radiation Dose (Gy)	Post-Irradiation Interval	Mode of Radiation	Radiation Types	Radiation-Impacted Pathways	Ref.
C57BL/6 mice	6–8 weeks	Jejunum	9	1 h	TBI	γ-rays	Proteasome and protein processing in the endoplasmic reticulum	[32]
CD2F1 mice	6–8 weeks	Jejunum	11	24 h	TBI	γ-rays	Rho family GTPases Signaling, glycolysis I, xenobiotic metabolism, 14-3-3-mediated signaling, and retinol biosynthesis	[35]
C57BL/6J mice	8–10 weeks	Ileum	8, 10, 12 and 14	1, 3, and 6 days	TBI	X-rays	Protein kinase A, acute phase response, and LXR/RXR signaling	[36]
C57BL/6J mice	6–8 weeks	Small intestines	13	3 days	ABI	γ-rays	DNA damage and apoptosis signaling	[38]
Sprague–Dawley rats	Adult	Ileum	10	4 days	ABI	γ-rays	FAS and glycolysis signaling pathway	[39]
Sprague–Dawley rats	6 weeks	Large intestines	20	10 weeks	ABI	X-rays	Complement and coagulation cascades, amoebiasis, phagosome, lysosome, focal adhesion, proteoglycans in cancer, and oxytocin signaling	[40]
Nonhuman primate (*Macaca mulatta*)	Adult	Jejunum	12	4, 8/9, 11/12, 15, and 21/22 days	PBI	X-rays	GP6 Signaling Pathway, acute phase response signaling, LXR/RXR pathway, and intrinsic prothrombin activation pathway	[41]

## Data Availability

Not applicable.

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
