# Peer review of "Radiation-Induced Intestinal Normal Tissue Toxicity: Implications for Altered Proteome Profile"

_genes, 2022, doi:10.3390/genes13112006_

Round 1

Reviewer 1 Report

In this manuscript the authors reviewed the intestinal tissue toxicity of ionizing radiation on the aspect of proteomic alterations. In general, although this article is informative, it is a descriptive review article in which the data and results from the cited papers were listed. Suitable judgment on these cited data and inductive views are required, especially the links with the tissue damage and pathological processing and changes (to present in form of a diagram), to further sort out the valuable information regarding the molecular signaling pathways or networks related with the radiation-induced intestinal toxicities, such as the early or later reactions associated signaling pathway(s).

Author Response

Reviewer 1

In this manuscript the authors reviewed the intestinal tissue toxicity of ionizing radiation on the aspect of proteomic alterations. In general, although this article is informative, it is a descriptive review article in which the data and results from the cited papers were listed. Suitable judgment on these cited data and inductive views are required, especially the links with the tissue damage and pathological processing and changes (to present in form of a diagram), to further sort out the valuable information regarding the molecular signaling pathways or networks related with the radiation-induced intestinal toxicities, such as the early or later reactions associated signaling pathway(s).

Response: This comment shows astute observation. Indeed, determination of altered pathways due to acute or delayed effects of radiation and whether these effects are dose dependent, would be of great value. A myriad of variables such as radiation dose and model, animal species and strain etc., can differentially alter signaling pathways. We have been unable to find any literature reference for a systematic study for alterations of signaling pathways as a result of neither early nor late reactions. However, two separate studies using a preclinical rat model have shown that acute reaction (4-day post-irradiation) altered both FAS and glycolysis signaling pathways, while delayed reactions (10 weeks following irradiation) mainly affect complement and coagulation cascades. Notably, an abdominal dose of 10 Gy g-rays was used for acute study and 20 Gy X-rays was used for delayed study. We chose to not highlight the links between differentially altered signaling pathways, and acute and delayed reactions of radiation due to the scarcity of literature and the complexity of this subject. Nonetheless, we added a statement in the conclusion section to emphasize the value of this information. 

Reviewer 2 Report

Radiation therapy is the most commonly applied approach to treatment for different types of cancer. However, gastrointestinal radiation injury is issue to limit the radiation dose in this treatment modality. In this review paper, authors summarized proteome analysis using animal model to understand side effects of radiotherapy in intestinal tissue. This reviewer’s comments are described below.

Major issues

1.     Authors should add item about age at exposure and tissues types of samples in table 1 and 2.

2.     There is no description of explanation for figure 1 in the text.

3.     Single-dose is used in most studies in table1 and 2. Dose response of biomarkers is unclear. Are there any biomarkers which indicate that change in their protein expression pattern seems to be related to radiation dosage?

Author Response

Reviewer 2

Radiation therapy is the most commonly applied approach to treatment for different types of cancer. However, gastrointestinal radiation injury is issue to limit the radiation dose in this treatment modality. In this review paper, authors summarized proteome analysis using animal model to understand side effects of radiotherapy in intestinal tissue. This reviewer’s comments are described below.

Major issues

  1. Authors should add item about age at exposure and tissues types of samples in table 1 and 2.

Response: We have added age at exposure and tissue type (specifically the part of the intestine) of the samples in Table 1 and 2 as suggested by the reviewer.

  1. There is no description of explanation for figure 1 in the text.

Response: We agree with the reviewer. We have added an explanation for Figure 1 in the revised version.

  1. Single-dose is used in most studies in table1 and 2. Dose response of biomarkers is unclear. Are there any biomarkers which indicate thatchange in their protein expression pattern seems to be related to radiation dosage?

Response: We would like to thank the reviewer for this comment. In this current review, one study showed that there is dose-dependent increase in 5 proteins (such as FABP1, FGA, FGB, FGG, and HP) in the intestinal sample of C57BL/6 male mice on day 3 following exposure to 8, 10, 12, and 14 Gy TBI. We have added a sentence on proteomics-based radiation biodosimetry in our revised version.